# Identifying the Antecedents of University Students' Usage Behaviour of Fitness Apps

**Jo-Hung Yu [1], Gordon Chih-Ming Ku [2,*], Yu-Chih Lo [3], Che-Hsiu Chen [4] and Chin-Hsien Hsu [3,*]**

1   Department of Marine Leisure Management, National Kaohsiung University of Science and Technology, Kaohsiung 811213, Taiwan; henry@nkust.edu.tw
2   Department of Sport Management, National Taiwan University of Sport, Taichung 404401, Taiwan
3   Department of Leisure Industry Management, National Chin-Yi University of Technology, Taichung 411030, Taiwan; loyuchih@ncut.edu.tw
4   Department of Sports Performance, National Taiwan University of Sport, Taichung 404401, Taiwan; jakic1114@ntus.edu.tw
*   Correspondence: GordonKu@gm.ntus.edu.tw (G.C.-M.K.); hsu6292000@yahoo.com.tw (C.-H.H.)

**Abstract:** The purpose of the study is to explore the antecedents of university students' fitness application usage behaviours by combining the theory of planned behaviour and the technology acceptance model. An anonymous questionnaire survey was adopted to address the objectives of the study. Purposive and snowball sampling was used to select eligible students from six universities in Zhanjiang City. An online survey was used to collect data from 634 eligible subjects, and partial least squares structural equation modelling was used to analyse the collected data. The results indicated that the students' perceived usefulness ($\beta = 0.17$, $p < 0.05$) and perceived ease of use ($\beta = 0.32$, $p < 0.05$) concerning the application and their attitude ($\beta = 0.31$, $p < 0.05$) toward it significantly influenced their usage intentions. Furthermore, perceived usefulness ($\beta = 0.11$, $p < 0.05$) and perceived ease of use ($\beta = 0.38$, $p < 0.05$) fully mediated the relationship between subjective norms and usage intentions. However, subjective norms and perceived behavioural control did not enhance the students' intentions to use fitness applications. That is, students' attitudes and fitness application design are the determinants of usage intention. Accordingly, improving students' fitness applications usage intention requires strategies that involve customised services, social networking, and collaboration with schools; this would further increase students' engagement in physical exercise.

**Keywords:** behavioural model; fitness apps; partial least squares structural equation modelling; technology acceptance model; theory of planned behaviour; behavioural intentions

## 1. Introduction

Mobile fitness applications (fitness apps) utilise persuasive technology to help individuals enhance their physical activity. Millions of people use fitness apps to enhance their physical and mental health. According to Fitness Worldwide [1], there were 822.42 million fitness app users worldwide in 2019, and this number is expected to reach 987.63 million by 2024. Furthermore, the global fitness app market value reached 3.15 billion USD in 2019 and is expected to reach 10.9 billion USD in 7 years [2,3]. Fitness apps have become a trend in the global fitness industry, which has resulted in new patterns of workout behaviour.

Fitness apps have been widely adopted to improve exercise behaviour and health management, substantially facilitating the reduction of global obesity rates and healthcare costs [4]. In contrast to traditional methods of exercise, fitness apps allow users to exercise anytime and anywhere with effective guidance and monitor, record, and manage their exercise process, thereby enhancing their exercise effectiveness. For example, Litman et al. [5] found that fitness apps users engaged in higher-intensity physical activity and showed greater self-efficacy and BMI control improvements than non-users. Therefore, research concerning the behaviour of fitness apps users has received widespread attention.

Many behavioural models and theories have been applied to fitness apps users' behaviour, such as the unified theory of acceptance and use of technology [6,7], prospect theory and self-efficacy theory [8], theory of planned behaviour (TPB) [9], technology acceptance model (TAM) [10], and theories based on persuasive technology [11]. Identifying the antecedents of fitness app users' behaviour would facilitate the development of strategic plans for business and marketing purposes and help formulate strategies to strengthen users' engagement in fitness apps [12]. However, a single theory has limited explanatory power; thus, combining multiple theories could improve the understanding of fitness apps users' behaviour.

TPB and TAM have been identified as rigorous foundations for explaining why people adopt a certain type of technology [13]. Combining these two theories has been applied in various fields, such as internet banking [12,14], to explain users' intentions of adopting new technologies and systems and continuing their use. It has been proposed that combining these two theories could better explain users' technology usage intention than a single theory [15]. Moreover, it has been argued that using only TAM to explain people's intention to use technology may oversimplify its effect on user behaviour and attitude. In contrast, TPB alone cannot uncover the factors that affect people's attitudes toward technology use [15]. Since information and communications technology development, fitness apps have been popularly used to support people's exercise needs [13]. Therefore, to satisfy fitness apps users' requirements, understanding their usage behaviour is necessary to design appropriate functions and provide supported service. TPB and TAM can offer a picture to comprehend the users' fitness apps usage behaviour.

Teenagers are major mobile application users in most societies, especially university students [16,17]. In Mainland China, teenagers, including university students, who are overweight or obese increased from 6.2 million in 1985 to 35.0 million in 2014. The government expects obese teenagers will reach 49.5 million in 2030 [18]. Fitness Apps have been regarded as an extrinsic motivation that enhances users' physical activity and improves their health [19]. Therefore, fitness apps are popular among university students for improving their health circumstances and supporting their exercise needs [16]. The issue of fitness apps benefits in physical and mental health has been widely explored in previous studies [20,21]. However, to our knowledge, no studies to date have identified the antecedents of university students' behaviour while using fitness apps; therefore, their usage behaviour model remains unclear. Accordingly, this study integrated TPB and TAM to investigate university students' intentions to use fitness apps and examine the behaviour underlying the use of fitness apps. Consequently, the findings could be made regarding how to promote fitness apps to university students and directly help resolve the obesity issue for teenagers in China.

## 2. The Present Study

TPB and TAM have been used to explain users' intentions and behaviour related to the use of new technologies and systems [22,23]. TPB and TAM are derived from the theory of reasoned action (TRA) proposed by [24]. TRA is a general framework used to predict human behaviour. It posits that nearly all human behaviour is based on the importance of personal beliefs. Individuals' attitudes and subjective norms are critical antecedents of behavioural intention, which is the main predictor of actual behaviour. TRA hypothesises that the more positive an individual's attitude and the greater the perception of subjective norms, the more likely a behavioural intention is generated; the stronger the behavioural intention, the more likely it is to result in actual behaviour.

To further understand individual behaviour, Ajzen [25,26] and Ajzen and Madden [27] developed TPB based on Bandura's concept of self-efficacy. The TPB framework consists of three functional concepts that affect behavioural intentions: attitudes, subjective norms, and perceived behavioural control. Within the TPB framework, the concept of attitudes refers to an individual's persistent tendency to evaluate a particular object or idea positively or negatively, which can be used to predict possible behaviours. Subjective norms denote

the social pressure perceived by an individual from significant others or relevant social groups that influences a given behavioural performance. Perceived behavioural control is an individual's perception of his or her ability to control the resources and opportunities needed to engage in a behaviour [28,29]. An individual with a positive attitude, sufficient support from the subjective norms of others, and strong perceived behavioural control will exhibit a solid intention to perform a certain behaviour. Furthermore, usage intentions and perceived behavioural control have a direct impact on behavioural performance. TPB has been applied to explore usage behaviour related to mobile apps, including mobile shopping [30], mobile learning [28], and smart home services [29]. Studies have shown that TPB has significant explanatory power concerning behavioural intentions toward using mobile apps [31].

TAM is another model used to explain technology adoption behaviours and understand the behavioural intentions of individuals using technology to accomplish tasks [32,33]. The TAM framework is predominately based on two functional concepts: perceived usefulness and perceived ease of use, which predict users' attitudes, behavioural intentions, and actual behaviour with respect to new technologies [10,32]. Perceived usefulness refers to the individual's belief that a particular technology or system can help them complete tasks; this represents the effectiveness of technology use. Perceived ease of use denotes the ease with which individuals use a technology or system to complete tasks. These perceptions can trigger users' positive attitudes and usage intentions toward technology. TAM and TPB have been combined (TAM–TPB model) to explain technology usage behaviour in various fields comprehensively. Cheng [31] demonstrated that the TAM–TPB model provides greater explanatory power than either of these theories in isolation. Therefore, the TAM–TPB model can provide a more comprehensive understanding of people's technology usage behaviours [12–15].

The TAM–TPB model may lead to different results when applied to different issues regarding technology usage behaviours. For instance, Hassan et al. [14] reviewed 23 studies in which the TAM–TPB model was applied to technology usage behaviours. The review indicated that the explanatory power of the TAM–TPB model ranged from 24.1% to 76%, with diversity in the strength of causal relationships among variables. The TAM–TPB model can reflect different results of technology usage behaviours in diverse fields and variation in participants' perspectives, and the model can provide precise and valuable information for designing technology functions that address customers' needs [15,34]. Therefore, applying the TAM–TPB model to university students' fitness apps usage behaviour could provide insight into their usage needs.

Previous studies of technology usage behaviours using the TAM–TPB model suggest that perceived usefulness and perceived ease of use are cognitive responses that reflect users' attitudes toward technology [35]. Thus, the users' attitudes toward technological products depend on whether the product is useful and easy to use. In addition, perceived usefulness and perceived ease of use also have a significant impact on intentions to use technological products [14,31,36]. For example, university students would likely display positive attitudes toward using fitness apps and high usage intentions if the apps were to exhibit high levels of perceived usefulness and perceived ease of use. Accordingly, the following two hypotheses were proposed:

**Hypothesis H1.** *The extent to which university students perceive a fitness app as useful and easy to use will significantly predict their attitude towards fitness apps.*

**Hypothesis H2.** *The extent to which university students perceive a fitness app as useful and easy to use will significantly predict their usage intentions.*

Perceived ease of use positively influences perceived usefulness and indirectly increases intentions to use technology [37]. For example, university students would regard a fitness app as a useful technology for achieving their goals if they perceived the app as

easy to use; this would indirectly increase university students' intentions to use the app. Therefore, the following two hypotheses were proposed:

**Hypothesis H3.** *The extent to which university students perceive a fitness app as easy to use will significantly predict the extent to which they perceive the app as useful.*

**Hypothesis H4.** *Perceived ease of use will indirectly predict university students' intentions to use a fitness app via perceived usefulness.*

Previous studies have widely reported that technology users' attitudes affect their usage intentions; a positive attitude toward technology increases users' technology usage intentions [38]. Moreover, subjective norms and perceived behavioural control are key antecedents that trigger users' technology usage intentions, while perceived behavioural control can directly enhance users' behavioural performance during technology usage [15,23]. University students will have high intentions to use a fitness app when they feel pressured by influential social groups whose members use fitness apps to exercise and perceive that a fitness app will help them direct their fitness activity. Therefore, high usage intentions and perceived behavioural control will increase university students' fitness apps usage behaviour. Accordingly, the following two hypotheses were proposed:

**Hypothesis H5.** *University students' attitudes, subjective norms, and perceived behavioural control will significantly predict their fitness app usage intentions.*

**Hypothesis H6.** *University students' perceived behavioural control and usage intentions will significantly predict their fitness app usage behaviour.*

Subjective norms engender particular behaviours and motivations; these norms are essential antecedents of technology users' perceived usefulness and perceived ease of use. Furthermore, perceptions of the usefulness and ease of use regarding technology are affected by the guidance and influence of significant social groups. For example, teachers who use fitness apps in physical education classes could foster students' attitudes that such apps are useful and easy to use, thereby further influencing students' intentions to use such apps [39,40]. Thus, subjective norms can be considered an antecedent of university students' perceived usefulness and perceived ease of using fitness apps; these norms indirectly affect the individual's fitness apps usage intentions. Accordingly, the final two hypotheses were proposed:

**Hypothesis H7.** *Subjective norms will significantly predict the extent to which university students perceive fitness apps as useful and easy to use.*

**Hypothesis H8.** *Subjective norms will indirectly predict university students' fitness app usage intentions via perceived usefulness and perceived ease of use of fitness apps.*

Previous studies using the TAM–TPB model proposed eight hypotheses and established a TAM–TPB model of university students' fitness apps usage (Figure 1) to identify the antecedents of their fitness apps usage behaviour.

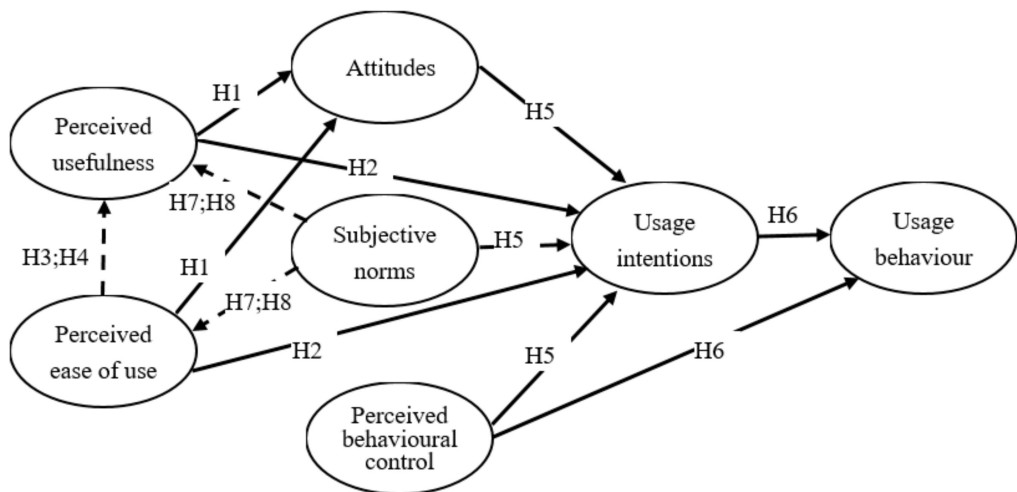

**Figure 1.** TAM–TPB model of fitness apps usage behaviour. Note: H8 and H4 are indirect predictions.

### 3. Methodology

*3.1. Research Design*

A questionnaire survey was conducted to investigate fitness apps usage behaviour of college students in Zhanjiang City, Guangdong Province, China. To enhance college students' physical and mental health, fitness apps have been used in mainland China to improve students' physical activity performance and increase their exercise frequency. For example, the LePao fitness app has become popular on university campuses in mainland China. The app supports physical education and records students' physical activity performance. As with Keep, Yodo Run, Cuckoo, etc., the other apps are also popular with the youth. Therefore, fitness apps may be considered an essential technology in Chinese students' campus life. Hence, college students in mainland China were selected as the respondents in this study.

A questionnaire was developed based on previous studies using TAM and TPB. Twenty college students were recruited to participate in the pre-test review of the questionnaire content to ensure the questionnaire was concise and improve the readability and comprehensibility of questions. Subsequently, two experts were invited to review the questionnaire: one professor and one associate professor in the Department of Social Sport with experience studying fitness technology. The formal questionnaire was constructed after two rounds of revision. Accordingly, the content validity of the questionnaire was established [41,42]. The questionnaire survey was conducted online. Invalid online questionnaires were deleted (e.g., questionnaires composed of identical answers and online questionnaires from the same intellectual property).

*3.2. Respondents and Procedure*

Purposive and snowball sampling were used to select eligible undergraduate students from six universities in Zhanjiang City, China, including freshmen, sophomores, juniors, and seniors. International students and postgraduates were excluded. Researchers asked teachers and staff working at the six universities to distribute the online questionnaires by WeChat. Before the online questionnaire distribution, researchers personally explained the purpose of this study and respondents' rights before the respondents provided informed consent. The WenJuan online questionnaire platform (https://www.wenjuan.com/ accessed on 20 June 2020) was selected to collect data. Respondents were also encouraged to invite acquaintances who were undergraduate students in Zhanjiang City to participate in the online questionnaire survey. The survey period was between 20 June 2020 and 2 August 2020. A total of 687 online questionnaires were returned, 53 of which were deleted because they consisted of identical answers and intellectual property. Finally, 634 valid online questionnaires were collected for an effective response rate of 92.3%.

### 3.3. Ethical Considerations

The present study was approved by the Tsaotun Pyschiatic Center Institutional Review Board for the ethical approval (IRB# 110031). A two-stage process was used to obtain respondents' informed consent. In the first stage, the teachers and staff who assisted with the online questionnaire distribution thoroughly explained the purpose of the study and respondents' rights. The respondents were permitted to complete the online questionnaire after they provided informed consent. In the second stage, the first page of the online questionnaire was a detailed informed consent form. Then, the respondents were asked to read the content to ascertain further that their rights were protected.

### 3.4. Measures

The survey questionnaire, which was developed based on TPB and TAM, consisted of two sections. The first section was a TPB–TAM instrument constructed according to Ajzen's [25,26] and Davis' [32] definitions of TPB and TAM. The scale comprised 24 items split into seven subscales: perceived usefulness, perceived ease of use, attitudes, subjective norms, perceived behaviour control, usage intentions, and usage behaviour (Table 1). Responses were collected via a five-point Likert scale, with options ranging from 'strongly disagree' (1) to 'strongly agree' (5). The second section of the questionnaire collected demographic data, including gender, grade, exercise frequency per week, duration of exercise per day, and usage frequency of fitness apps.

**Table 1.** The items of the scale.

| Variables | Items |
|---|---|
| Perceived usefulness | The fitness apps that I use can<br>1.   Meet my exercise needs (PU1)<br>2.   Provide suggestions for exercise (PU2)<br>3.   Improve my exercise efficiency (PU3)<br>4.   Lean valid exercise (PU4) |
| Perceived ease of use | The fitness apps that I use<br>5.   Are easy to operate (PEU1)<br>6.   Have a simple interface (PEU2)<br>7.   Able to check in any time (PEU3) |
| Attitudes | The fitness apps that I use are<br>8.   More effective than traditional exercise (A1)<br>9.   More convenient than traditional exercise (A2)<br>10.  Exceeded my expectations (A3) |
| Subjective norms | I use the fitness apps because<br>11.  I am influenced by others' usage (SN1)<br>12.  They were recommended by friends or peers (SN2)<br>13.  Used by people in my social world (SN3) |
| Perceived behaviour control | The fitness apps that I use<br>14.  Achieve the goal (PBC1)<br>15.  Complete the required exercise intensity (PBC2)<br>16.  Perform the exercise (PBC3)<br>17.  Accomplish the training task (PBC4) |
| Usage intentions | I will<br>18.  Recommend the fitness apps to my friends (UI1)<br>19.  Use the fitness apps every time (UI2)<br>20.  Continue to use the fitness apps (UI3) |

**Table 1.** *Cont.*

| Variables | Items |
|---|---|
| Usage behaviour | I have<br>21.   Recommended the fitness apps to my friends (UB1)<br>22.   Used the fitness apps every time (UB2)<br>23.   Been continued using the fitness apps (UB3) |

### 3.5. Data Analysis

SPSS 18.0 and SmartPLS 3.0 statistical software were used to analyse data. Descriptive statistics were used to analyse the respondents' demographic information and further understand the survey distribution. In addition, partial least squares structural equation modelling (PLS-SEM) was used to analyse the reliability and validity of the scale (measurement model) and to verify the TAM–TPB model as applied to university students' fitness apps usage behaviour (structural model). Finally, via PLS-SEM, the study provided a comprehended picture to understand the relationship between antecedents and fitness apps usage behaviour from students' perspectives.

## 4. Results

### 4.1. Demographic

The respondents' demographic (Table 2) indicated that most respondents are female (65.0%). More than half of the respondents are sophomores (53.5%). Approximately one-third of the respondents exercise more than four times per week (30.4%). The most frequent duration of exercise per day is less than 60 min (55.0%). Most respondents use fitness apps occasionally (75.1%).

**Table 2.** Respondents' demographic.

| Variable | Frequency | Percentage | Variable | Frequency | Percentage |
|---|---|---|---|---|---|
| **Gender** | | | **Duration of Exercise** | | |
| Male | 222 | 35.0% | Less than 60 min | 349 | 55.0% |
| Female | 412 | 65.0% | 61–120 min | 230 | 36.3% |
| **Degree** | | | 121–180 min | 47 | 7.4% |
| Freshman | 222 | 35.0% | More than 181 min | 8 | 1.3% |
| Sophomore | 339 | 53.5% | **Exercise Frequency per Week** | | |
| Junior | 28 | 4.4% | 1 time | 162 | 25.6% |
| Senior | 45 | 7.1% | 2 times | 128 | 20.2% |
| **Fitness App Usage** | | | 3 times | 151 | 23.8% |
| Occasionally | 476 | 75.1% | More than 4 times | 193 | 30.4% |
| Usually | 108 | 17.0% | | | |
| Always | 50 | 7.9% | | | |

### 4.2. Measurement Model Analysis

The reliability and validity test (Table 3) reveal that the factor loadings of the observed variables of the TPB–TAM model are above 0.70 (attitudes = 0.84–0.87, subjective norms = 0.82–0.84, perceived behavioural control = 0.81–0.87, perceived usefulness = 0.80–0.86, perceived ease of use = 0.79–0.85, usage intentions = 0.82–0.90, usage behaviour = 0.86–0.90), which indicates that the scale exhibited good convergent validity. The Cronbach's $\alpha$ coefficients of the latent variables exceed 0.70, and the composite reliability coefficients of the latent variables are higher than 0.80 (attitudes = 0.82 ($\alpha$)/0.89 (composite reliability), subjective norms = 0.77/0.87, perceived behavioural control = 0.87/0.91, perceived usefulness = 0.85/0.90, perceived ease of use = 0.76/0.86, usage intentions = 0.84/0.90, usage

behaviour = 086/0.91), which indicates that the TPB–TAM model is internally consistent. In addition, the average variance extracted (AVE) of the variables exceed 0.50 (attitudes = 0.73, subjective norms = 0.68, perceived behavioural control = 0.71, perceived usefulness = 0.69, perceived ease of use = 0.68, usage intentions = 0.76, usage behaviour = 0.78), which indicates that the variance in the latent variables are largely captured by the observed variables.

**Table 3.** Reliability and validity of TPB–TAM model.

| Variable | Item | Factor Loading | Cronbach's $\alpha$ | C.R. | AVE |
|---|---|---|---|---|---|
| Attitude | A1 | 0.86 | 0.82 | 0.89 | 0.73 |
| | A2 | 0.84 | | | |
| | A3 | 0.87 | | | |
| Subjective norm | SN1 | 0.84 | 0.77 | 0.87 | 0.68 |
| | SN2 | 0.82 | | | |
| | SN3 | 0.82 | | | |
| Perceived behavioural control | PBC1 | 0.81 | 0.87 | 0.91 | 0.71 |
| | PBC2 | 0.87 | | | |
| | PBC3 | 0.87 | | | |
| | PBC4 | 0.83 | | | |
| Perceived usefulness | PU1 | 0.81 | 0.85 | 0.90 | 0.69 |
| | PU2 | 0.82 | | | |
| | PU3 | 0.86 | | | |
| | PU4 | 0.80 | | | |
| Perceived ease of use | PEU1 | 0.85 | 0.76 | 0.86 | 0.68 |
| | PEU2 | 0.83 | | | |
| | PEU3 | 0.79 | | | |
| Usage intention | UI1 | 0.82 | 0.84 | 0.90 | 0.76 |
| | UI2 | 0.88 | | | |
| | UI3 | 0.90 | | | |
| Usage behaviour | UB1 | 0.90 | 0.86 | 0.91 | 0.78 |
| | UB2 | 0.88 | | | |
| | UB3 | 0.86 | | | |
| Goodness of Fit | | | | 0.41 | |

Note: A = attitude; SN = subjective norm; PBC = perceived behavioural control; PU = perceived usefulness; PEU = perceived ease of use; UI = usage intention; UB = usage behaviour; Cronbach's $\alpha$ = internal consistency coefficient; C.R. = composite reliability coefficient; AVE = average variance extracted.

Furthermore, goodness-of-fit (GoF) is a critical criterion for model fit in partial least squares analysis. The GoF value reflects the explanatory ability of observed data. In the following GoF formulation, the higher the GoF, the greater the explanatory power of the model's estimated parameters [43]:

$$\text{GoF} = \sqrt{\text{average AVE} \times \text{average } R^2} \tag{1}$$

Akter et al. [43] suggest that GoF values can be divided into three ranges: above 0.36 denotes a high level of model fit, 0.25–0.35 denotes a moderate level of model fit, and 0.10–0.24 denotes an acceptable level of model fit. Thus, the GoF value of the TPB–TAM model was 0.46, which reflects a good fit of the model in this study (Table 1).

Discriminant validity analysis indicates that the square root of the AVE of the latent variables is greater than the correlation between the latent variables in the TPB–TAM model (Table 4). The only pairwise correlation that exceeds the AVE value is that between usage intention and usage behaviour. Overall, the latent variables of the TPB–TAM model exhibit acceptable discriminant validity.

**Table 4.** Discriminant validity of TPB–TAM model.

|     | SN | UI | UB | PEU | PU | PBC | A |
|-----|------|------|------|------|------|------|------|
| **SN** | **0.83** | | | | | | |
| **UI** | 0.35 | **0.87** | | | | | |
| **UB** | 0.39 | 0.89 | **0.88** | | | | |
| **PEU** | 0.38 | 0.67 | 0.64 | **0.82** | | | |
| **PU** | 0.38 | 0.68 | 0.63 | 0.74 | **0.83** | | |
| **PBC** | 0.32 | 0.60 | 0.55 | 0.62 | 0.75 | **0.85** | |
| **A** | 0.37 | 0.65 | 0.65 | 0.54 | 0.66 | 0.62 | **0.86** |

Note: A = attitude; SN = subjective norm; PBC = perceived behavioural control; PU = perceived usefulness; PEU = perceived ease of use; UI = usage intention; UB = usage behaviour; numbers in bold = square root of AVE.

### 4.3. Structural Model Test

The results of the TPB–TAM structural model test (Figure 2) indicate that attitudes toward fitness apps are significantly influenced by its perceived usefulness ($\beta$ = 0.58, $p < 0.05$) and perceived ease of use ($\beta$ = 0.12; $p < 0.05$), as are usage intentions ($\beta$ = 0.17, $p < 0.05$ and $\beta$ = 0.32, $p < 0.05$, respectively). University students' attitudes toward fitness apps are enhanced if they believe that fitness apps could effectively support their fitness goals and are easy to use. In addition, perceived ease of use is significantly associated with perceived usefulness ($\beta$ = 0.70, $p < 0.05$). That is, an easier-to-use interface design is associated with the greater perceived usefulness of fitness apps. Accordingly, perceived usefulness and perceived ease of use are two critical external variables that influence the students' attitudes and usage intentions regarding fitness apps.

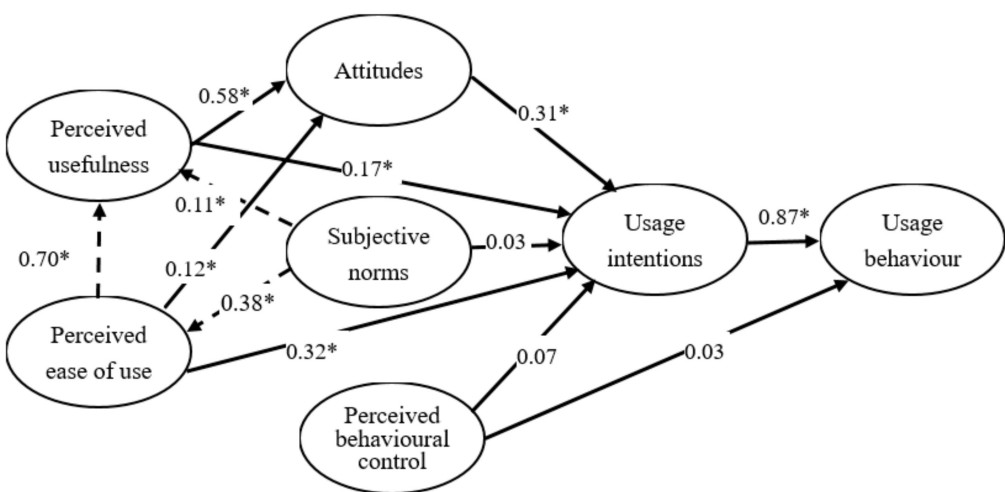

**Figure 2.** The TPB–TAM model. Note: * $p < 0.05$.

Surprisingly, students' fitness apps usage intentions are not associated with subjective norms ($\beta$ = 0.03, $p > 0.05$) or perceived behavioural control ($\beta$ = 0.07, $p > 0.05$). Attitudes are significantly associated with students' fitness apps usage intentions ($\beta$ = 0.31, $p < 0.05$). The students' intentions to use fitness apps are not affected by their acquaintances or the perception that the fitness training program is easier to follow. Furthermore, subjective norms significantly influences the extent to which the students' perceived fitness apps as useful ($\beta$ = 0.12, $p < 0.05$) and easy to use ($\beta$ = 0.38, $p < 0.05$). Students perceived fitness apps as easy to use if they are aware that their acquaintances perceived apps in this way. Finally, students' perceived behavioural control do not directly influence fitness apps usage behaviour ($\beta$ = 0.03; $p > 0.05$), whereas behavioural intentions strongly affect actual usage behaviour ($\beta$ = 0.87; $p < 0.05$). That is, usage intentions are critical for increasing fitness apps usage. In contrast, easier control of the training program is an insignificant attraction in terms of increasing usage.

There are further indirect effects among variables. Perceived usefulness and perceived ease of use fully mediate the relationship between subjective norms and usage intentions. The students' fitness apps usage intentions are only affected by subjective norms through perceived usefulness and ease of use. To enhance their fitness apps usage intentions, the students' acquaintances perceiving such apps as useful and easy to use is critical. Furthermore, perceived usefulness partially mediates the relationship between perceived ease of use and usage intentions. Thus, students' fitness apps usage intentions are directly influenced by perceived ease of use or indirectly influenced via perceived usefulness. The TPB–TAM model achieves an explanatory power of 78.6%.

## 5. Discussion

This study integrates TPB and TAM as a combined model to explore university students' fitness apps usage behaviours. Of several vital findings, the first is that the perceived utility of the fitness program and more straightforward interface operation improve students' attitude toward fitness apps and increase their usage intentions; the latter can eventually increase actual fitness apps usage. These findings are consistent with [44], who found that perceived usefulness and ease of use ratings made by health management technology users positively influenced their attitudes and usage intentions and indirectly affected their usage behaviour. Once individuals believe that technology can effectively assist them in achieving their goals and the system is simple to operate, perceived usefulness and ease of use become triggers that drive individuals to adopt technological products [29,45]. Therefore, fitness app design for students should provide appropriate workouts via a simple interface [14].

This study also found that students' attitudes influence their fitness apps usage intentions. This finding is consistent with most TPB studies of technology usage. Users' perceptions and evaluations of technological products determine usage intentions [45]. Accordingly, building a positive attitude toward fitness apps is essential to increase students' intentions to use fitness apps to direct their exercise behaviour. Fitness app companies can cooperate with schools, and schools can integrate fitness apps into physical education to enhance students' positive perceptions of fitness apps and encourage their usage habits. For example, Gowin et al. [16] integrating a fitness app with a health education curriculum in school can increase students' fitness apps usage frequency and improve their health status.

Subjective norms do not significantly affect students' fitness apps usage intentions. This result differs from most research findings related to the TPB–TAM model. Students' fitness apps usage intentions and behaviour are not influenced by social pressure and recommendations by their acquaintances, nor by more accessible training programs. Self-efficacy determines whether individuals are affected by social pressure [46]. Low self-efficacy leads individuals to succumb to social pressure. Accordingly, the students may exhibit higher self-efficacy concerning fitness training performance because the difficulty of the training program (perceived behaviour control) does not influence their usage intentions. Self-efficacy can be considered an external variable reassessed in future studies in terms of its role in the relationship between subjective norms and usage intentions.

Furthermore, students' perceived behavioural control does not influence their fitness apps usage intentions or behaviour. Therefore, fitness apps should satisfy students' goals and enrich their usage experience, an appropriate challenge. Flow experience may be a critical causal factor in students' engagement with fitness apps, which occurs when teenagers' skills and challenges achieve an optimal balance [47]. The option of customised fitness apps can help users set and achieve their ideal goals and increase challenges and improve the quality of exercise experiences [48], such as using big data to intelligently design the training program according to students' physical characteristics. Therefore, students can receive flow experiences from customised training programs offer by fitness apps, which can further enhance usage intentions and behaviour. Moreover, flow experiences may be vital in understanding student's fitness apps usage intentions and behaviour; these experiences are thus worth further investigation in future studies.

Subjective norms fully mediate the relationships between perceived usefulness, perceived ease of use, and usage intentions. Perceived usefulness partially mediates the relationship between perceived ease of use and usage intentions. The students are more willing to use a fitness app if they learn from others that the app is useful and easy to use; in particular, perceived usefulness is crucial in students' fitness apps usage intentions. How to convey a fitness app's usefulness and ease of use to students is critical to encourage them to use the app for exercise. Social networking related to a fitness app is one medium through which fitness apps usage experiences can be disseminated. Social networking comprises online platforms that permit individuals to share personal exercise experiences. This may motivate individuals to engage in fitness activities, which can help individuals using fitness apps to accomplish their goals; these persons may then further share their own experiences and achievements with other users of the social network platform [49]. For most students, sharing personal experiences via social networking is a part of their daily life. Therefore, to increase students' perceived usefulness and ease of use, social networking can be regarded as a promoted strategy, such as integrating social networking platforms with new social media or key opinion leaders (KOL).

The TPB–TAM model significantly explains university students' fitness apps usage behaviour, although subjective norms and perceived behaviour control do not affect their usage intention or behaviour. Furthermore, the combined model can indicate the different variables and paths that affect technology usage behaviour, which may help companies to design more user-aligned technological applications or systems [15,35]. Accordingly, the TPB–TAM model provides insights into the critical antecedents of university students' fitness apps usage behaviour.

## 6. Strengths, Limitations, and Future Research

The study utilises a rigorous research process to verify the TPB–TAM model of university students' fitness apps usage behaviour. However, there are two main limitations to the study. First, the respondents are students in China, which may explain students' fitness apps usage in this specific area. However, the TPB–TAM model can be applied to different areas, including western countries, to understand students' fitness apps usage behaviour further and compare their usage intentions and behaviour in the future study. Second, this study is cross-sectional and reflects students' usage intentions and behaviour at a specific period. Accordingly, there is a need for longitudinal research to elucidate students' fitness apps usage behaviour more fully.

## 7. Conclusions

Fitness apps are a vital technology to assist university students in improving and maintaining health. The current study adopts the TPB–TAM model to identify the antecedents of students' fitness apps usage behaviour. Perceived usefulness, perceived ease of use, and attitudes are the significant antecedents that affect students' fitness apps usage intentions. However, subjective norms and perceived behavioural control have no significant effects on students' fitness apps usage intention. Moreover, subjective norms mediate perceived usefulness, ease of use, and usage intention.

**Author Contributions:** Conceptualization, J.-H.Y., G.C.-M.K. and C.-H.H.; methodology, J.-H.Y. and G.C.-M.K.; formal analysis, J.-H.Y., G.C.-M.K. and Y.-C.L.; investigation, Y.-C.L. and C.-H.C.; resources, Y.-C.L. and C.-H.H.; data curation, Y.-C.L. and C.-H.C.; writing—original draft preparation, J.-H.Y. and G.C.-M.K.; writing—review and editing, G.C.-M.K. and C.-H.H.; project administration, Y.-C.L., C.-H.C. and C.-H.H. All authors have read and agreed to the published version of the manuscript.

**Funding:** This research received no external funding.

**Institutional Review Board Statement:** The study was conducted according to the guidelines of the Declaration of Helsinki, and approved by the Institutional Review Board of Tsaotun Psychiatric Center (IRB# 110031 and 30 July 2021).

**Informed Consent Statement:** In light of anonymous questionnaire survey in this study, the content of informed consent has been informed to respondents before starting data collection.

**Data Availability Statement:** Data are not accessible. Accordingly to the informed consent of the study, only the researchers can access the data. Therefore, data cannot be made publicly available.

**Conflicts of Interest:** The authors declare no conflict of interest.

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
