# Peer review of "Identifying the Antecedents of University Students’ Usage Behaviour of Fitness Apps"

_sustainability, doi:10.3390/su13169043_

Round 1

Reviewer 1 Report

Thank you for submitting to Sustainability. Doing research on apps and exercises is very meaningful research. In particular, from the perspective of the 4th industry, the connection between social science and exercise is meaningful. However, for better publication, I suggest the following:

Author Response

Dear Reviewer,

Thank you very much for forwarding to us the valuable comments. Your time and effort are very much appreciated. We have revised our manuscript based on your comments, and have summarized the changes in the attached file.

Reviewer 2 Report

Comments to the Authors

Thank you for the opportunity to review this paper. Overall, it is a well conducted and written study on an important topic. Please also see some concerns related to this paper.

Title

The title contains the key features of the article. Also, the title is attractive and might spot interest in the reader. However, I would ask authors to change “4.0” in the title as it is confusing.

Abstract

The abstract is well written and important information is provided for the reader. However, Author’s state two times that it was online survey, please change this. Also, please add some specific values to the abstract.  

Introduction

Overall, the introduction reads well. The authors provide adequate review of the existing literature.  However, I am not sure how should be the “1. Introduction” and “2. Literature review” different from each other? These two should not be separate global headings, it should be all introduction until “the present study” or “methods”. Still, you can use the subheadings if necessary.

Lines 74 to 76: Please specify which type of motivation are the Authors referring to. Based on Ryan and Deci (2020), there are extrinsic and intrinsic forms of motivation, and these forms of motivation are differently related to physical activity. Authors state that “fitness apps are widely used by teenagers to motivate physical activity”. It is important to note that previous research has highlighted it is specifically intrinsic motivation that is related to adolescents’ physical activity (Kalajas-Tilga et al., 2020), and not other forms of motivation.

Kalajas-Tilga, H., Koka, A., Hein, V., Tilga, H., & Raudsepp, L. (2020). Motivational processes in physical education and objectively measured physical activity among adolescents. Journal of Sport and Health Science, 9(5), 462−471. https://doi.org/10.1016/j.jshs.2019.06.001  

Ryan, R. M. & Deci, E. L. (2020). Intrinsic and extrinsic motivation from a self-determination theory perspective: Definitions, theory, practices, and future directions. Contemporary Educational Psychology, 61, 101860. https://doi.org/10.1016/j.cedpsych.2020.101860

Please make a separate section “The present study” in which you introduce all the hypotheses.

Hypotheses – please use “predicts” instead of “influence”. A cross-sectional study cannot identify which variable influences another variable.

Figure 1 – please provide a figure with better quality, currently it is very foggy. Also, please remove blue and red underlines, avoid breaking the lines with textboxes, and remove small grey arrows. Also, indirect hypotheses are not clear on the Figure – it is confusing when you present several H’s for the same arrows/lines.

Methodology

Overall, the methodology is clear.

3.5. Data analysis – I think this section could be more detailed. Several statistics could be shortly mentioned here which are reported in the results section.

Results and discussion

Where there is heading “Results and discussion”, if there is later a separate heading “Discussion”? I think first it should be only “Results”.

The heading of the Table 2 could be improved. Currently it is too vague and confusing. Also, I am not sure if the short names of items (e.g., PBC1) has any meaning to the reader. These should be linked with specific items of the scales.

Lines 289 to 296: this should be presented in the data analysis section.

Table 3 – please be more specific with the headings.

4.3. Structural Model Test – please provide more accurate p-values.

Figure 2 – please provide a figure that has better quality, remove red underlines, remove small grey arrows, and please do not let textboxes cut arrows/lines.

Discussion

Overall, discussion is very well written but could benefit from more depth interpretation of the results. Specifically, I encourage Authors to find more recent studies in this field of topic and propose suggestions for the future research. For example, the current study relied on self-reported data. How could future research measure the behaviour more objectively?

Please provide a separate section “Strengths, limitations and future research”.

Author Response

(The authors gave the same response as above.)

Reviewer 3 Report

This paper has made a good effort to combine a well tested theory (TPB) with a new model (TAM) providing preliminary support for this hybrid model as applied to teenagers usage of Fitness apps. Results may apply to both further research and app industry to create or improve fitness apps, which contribute significantly to the promotion of physical activity and public health. The paper is very well written, with a clear structure, appropriate methodology,  and a discussion to the point. I have no comments for improvement.   

Author Response

Dear Reviewer,

Thank you very much for forwarding to us the valuable comments. Your time and effort are very much appreciated. 

Round 2

Reviewer 1 Report

I hope that the results of the abstract can be supplemented a little more. It would be good to present a number or a p-value even if it is a part of the most representative result.

Author Response

Dear Reviewer,

Thank you again for forwarding to us the valuable comments. Your time and effort are very much appreciated. We have revised our manuscript based on your comments, and have summarized the changes in the attached file.

Reviewer 2 Report

Authors have done a great job addressing all the issues. My only concern is that please do not use questions in the section of "6. Strengths, limitations and future research" to propose suggestions for future research. Instead, please provide more specific directions for future research that is based on the literature. 

Author Response

(The authors gave the same response as above.)
